# Surface Acoustic Wave Biosensor with Laser-Deposited Gold Layer Having Controlled Porosity

**Dana Miu, Izabela Constantinoiu** **, Valentina Dinca** **and Cristian Viespe** *

Laser Department, National Institute for Laser, Plasma and Radiation Physics, Atomistilor 409, 077125 Magurele, Romania; dana.miu@inflpr.ro (D.M.); izabela.constantinoiu@inflpr.ro (I.C.); valentina.dinca@inflpr.ro (V.D.)
* Correspondence: cristian.viespe@inflpr.ro

**Abstract:** Laser-deposited gold immobilization layers having different porosities were incorporated into love wave surface acoustic wave sensors (LW-SAWs). Variation of pulsed laser deposition parameters allows good control of the gold film morphology. Biosensors with various gold film porosities were tested using the biotin–avidin reaction. Control of the Au layer morphology is important since the biotin and avidin layer morphologies closely follow that of the gold. The response of the sensors to biotin/avidin, which is a good indicator of biosensor performance, is improved when the gold layer has increased porosity. Given the sizes of the proteins, the laser-deposited porous gold interfaces have optimal pore dimensions to ensure protein stability.

**Keywords:** SAW sensor; pulsed laser deposition; biosensor; Au; nanoporous film; love wave



## 1. Introduction

There is currently great interest in the development of sensitive, portable, low-cost, miniaturized biosensing devices with rapid response [1–3]. A variety of biosensors have been investigated for the detection of biological analytes such as nucleic acids, proteins, or antibodies. Optical biosensors based on surface plasmon response (SPR) or surface-enhanced Raman spectroscopy (SERS) have very good sensitivity but require additional instrumentation, making remote operation and readout difficult [4–6]. Bulk acoustic wave (BAW) sensors allow portability but have lower sensitivity due to the relatively low frequency at which they operate [1,7].

Surface acoustic wave (SAW) sensor operation is based on the perturbation of the propagation of the SAW in the presence of an analyte, primarily through mechanical or acoustoelectric effects [8,9]. Due to the fact that wave propagation is limited to the surface, SAW sensors are inherently very sensitive to changes at this surface, more so, for example, than bulk acoustic wave sensors [1,8]. However, SAW sensors have excessive damping losses in liquids, which limits their use in biosensors [1,10]. Modification of the SAW structure through the incorporation of a guiding layer prevents penetration of the wave into the liquid, thereby avoiding excessive damping. This allows sensor operation in a liquid, as required for most biosensors [11,12]. SAW sensors having a love wave structure (LW-SAW) are comparable in sensitivity and reliability to other detection methods such as chromatography–mass spectrometry or ELISA [1]. In addition, they allow label-free recognition of analytes, i.e., recognition of analytes without the use of external reagents or chemicals. One of the ways of improving biosensor sensitivity is to increase the sensor active area [13].

Due to its biocompatibility and chemical stability, gold is widely used in biosensors. In particular, nanoporous gold has been successfully used in certain types of sensors, such as electrochemical sensors, amperometric, or quartz crystal microbalance (QCM) sensors, increasing their sensitivity through a high surface/volume ratio [14]. Nanoporous gold has been used in a QCM, where it proved to increase the gas sensitivity 40 times, compared

to a dense gold layer [15]. However, although porous sensitive films have been proven to increase the sensitivity of SAW gas sensors, the use of nanoporous gold in LW-SAW for detecting biological materials has not, to our best knowledge, been reported. In research previously reported, we have studied the effect of gold layer porosity on the response of SAW sensors to chloroform [16]. Gold layers with various porosities were deposited on top of a polymer guiding layer, and the acetylcholinesterase (AChE) enzyme was deposited on top of the gold layer as a sensitive layer. To the best of our knowledge, this was the first time that the use of a nanoporous gold layer was reported in a SAW-type sensor. The porosity of the underlying gold layer proved to influence the morphology of the AChE layer, and thereby the response of the LW-SAW sensor to chloroform. Important sensing properties such as sensitivity and limit of detection (LOD) were improved considerably in the case of porous gold layers, in comparison to a conventional dense gold layer. The nature of the gold deposited onto a substrate can be controlled from separate nanoparticles (NPs) to NP agglomerations, island growth, or compact thin films by using a pulsed laser deposition (PLD) technique, through variation of deposition parameters such as pressure, temperature or target-substrate distance [16–18]. Since the design of the sensor was the same as that for an LW-SAW biosensor, we considered that the results obtained in [16] can be used to improve the properties of SAW-based biosensors, by controlling the nanostructure of the gold immobilization layer, in combination with other enzymes or proteins.

Among the most used biosystems based on proteins within biosensing applications, the biotin–avidin coupling represents a well-known standard model used for labeled immunosensors to detect specific biomarkers for diseases such as cancer and influenza; it is also used for a variety of biological structures and processes since it has proven useful in the detection and localization of biological materials such as antigens, glycoconjugates, or nucleic acids [2,19].

Moreover, given the fact that porous gold interfaces for sensors must be designed in order to ensure that the pore size is correlated to the protein size, thus allowing proteins to attach to the surface and diffuse inside the pore openings, the present paper reports results obtained for LW-SAW sensors based on gold layers with different porosities obtained by PLD. The porosities of the sensors' active area were designed for testing as a proof of concept for biosensors using the biotin–avidin reaction.

## 2. Materials and Methods

The LW-SAW sensor is a delay-line type operating at a frequency of 69 MHz [20]. The 0.5 mm thick piezoelectric quartz crystal substrate (Roditi International Corporation Ltd.; London, UK) is Y-cut (42.75°), with a propagation direction of 90° with respect to the x-axis, is cut in parallelogram geometry (in order to reduce unwanted SAW reflections). The interdigital transducers (IDTs) consist of 50 input and output electrode pairs, with an aperture width of 2500 μm, and a wavelength of about 45 μm. The IDT Cr and Au metallization films, fabricated using photolithography, have thicknesses of 10 and 150 nm, respectively. The input IDTs convert the electrical signal into a surface acoustic wave, and the output IDTs convert it back into an electrical signal. The changes in the electrical signal detected at the output of the IDTs result from the interaction at the level of the surface of the sensors. An image of the sensor is given in Figure 1.

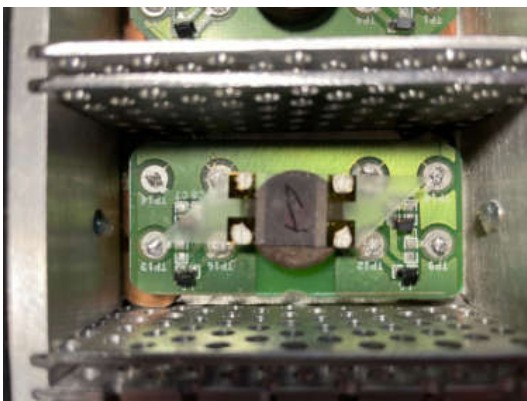

**Figure 1.** Photograph of the LW-SAW sensor mounted for measurement.

Polymethylmethacrylate (PMMA) is often used as a guiding layer in LW-SAW sensors due to its good mechanical and elastic properties, including low shear acoustic velocity (1105 m/s), relatively low density (1.17 g/cm³), and high stiffness module (1.7 GPa) [8,20,21]. PMMA also protects the interdigital electrodes from the various chemicals used in subsequent functionalization steps and avoids the well-known problem of poor adhesion of gold onto quartz. In our case, the PMMA guiding layer (Micro Resist Technology GmbH, Berlin, Germany) was applied by spin coating over the sensor surface after cleaning using the following successive steps: 100 rps—5 s; 500 rps—5 s; 1500 rps—20 s; 2500 rps—40 s; 1500 rps—20 s; 500 rps—20 s. This resulted in a polymer layer with uniform thickness over the entire sensor surface. The deposition was immediately followed by gradual heating of the sensor to 190 °C in one hour, leading to solidification of the polymer and resulting in a layer 2 μm thick.

The gold layers were deposited onto PMMA by PLD in various conditions in order to control their morphology and porosity. A laser with ns pulse durations (Nd-YAG EKSPLA model NL301HT) was used at 532 nm emission wavelength, at a repetition rate of 10 Hz, using an energy per pulse of about 75 mJ. The gold target (99.995% purity) was placed 4 cm away from the substrates. Gold films were deposited directly onto Si (100) substrates for SEM analysis. Depositions were made in the area between the input and output IDTs, on an $8 \times 10$ mm² surface using a mask, in order to avoid affecting the electrodes. For AFM analysis of the film morphology and for measurement of LW-SAW sensor properties, Au films were deposited onto the SAW sensor surfaces described above, on top of the PMMA layer. The deposition was made onto substrates at room temperature for all the results presented here. The morphology and porosity of the films were controlled through variations of the pressure in the deposition chamber and the number of laser ablation pulses. In the present experiments, the number of ablation pulses was varied between 1000 and 50,000. The pressure in the deposition chamber was varied between $10^{-5}$ Torr and 4 Torr Ar using a system for the control of pressure and gas flow. More details on the deposition system are given in [16].

The biotin and avidin proteins purchased from Sigma-Aldrich were used as received and dissolved in phosphate-buffered saline (PBS) (pH ~7.4) to a concentration of 0.01 mg/mL. The gold layers were functionalized with avidin and, respectively, avidin–biotin, which were successively incubated onto the gold layers in 12 h incubation sequence in a moisture chamber at 4 °C. Avidin functionalization step was followed by blocking step with bovine serum albumin (BSA) solution (a concentration of 0.01 mg/mL) for 2 h. Each step was followed by a washing cycle consisting of 2 times PBS washing and 2 times distilled water washing for removing any unbound protein and salt crystals. Nitrogen gas was used for drying the samples after each washing cycle. AFM was used to analyze the morphology of the surface after the deposition of each layer.

Analysis of the surface of the nanostructured gold films deposited in various conditions onto Si substrates was made by SEM (Thermoscientific Apreo S, Waltham, MA, USA).

The surface of the various layers that make up the sensor (PMMA, gold, avidin, biotin) were investigated by AFM (Park System XE-100, Suwon, Korea) after each successive deposition step.

The frequency changes of the sensors were monitored using a frequency counter (Pendulum CNT-91, Spectracom Corp, Rochester, NY, USA) connected to a computer with Time View III software (Pendulum Instruments, Banino, Poland). A DHPVA-200 FEMTO amplifier (Messtechnik GmbH, Berlin, Germany) was used to compensate for the signal loss from the oscillating circuit. All measurements were made in a room where the temperature and humidity were kept constant at 29 °C and the humidity at 41%. The sensors were allowed to thermalize to room temperature, and then measurements were made. The noise level was measured in air (without analyte) after about 30 min operation of the sensor setup for thermalization. The noise level, estimated by measuring the frequency fluctuation over 10 min, represents the maximum frequency deviation from the trend line. Thus, a noise level between 120 Hz and 140 Hz was obtained for all the sensors. The noise level is much less than the frequency shift of the sensors (between 4 and 19 kHz). Four sensors containing gold layers with different morphologies were functionalized with avidin and biotin. The frequency shift caused by avidin deposition onto the gold surface and the shift caused by functionalization of biotin onto avidin, respectively, were determined. One sensor (S1) contained a dense Au layer deposited at low pressure ($10^{-5}$ Torr). The other three sensors had different gold layer morphologies and porosities obtained by depositing the gold at different Ar pressures and pulse numbers: S2, 1 Torr Ar and 10,000 pulses; S3, 4 Torr and 10,000 pulses; S4, 4 Torr, and 30,000 pulses.

## 3. Results

### 3.1. Morphology

AFM permitted obtaining images of the various SAW sensor without affecting the polymer or protein layers. The surface of the 2 μm thick PMMA layer deposited onto quartz, as described above, is relatively smooth, with a roughness of about 0.3 nm (Figure 2a). Deposition of a gold film in vacuum conditions (pressure < $10^{-5}$ Torr) on top of the polymer produces characteristic wrinkling of the polymer surface due to interaction with the incident high energy gold species [22]. The wrinkling appears since deposition on top of polymer surfaces leads, in the case of inorganic materials with high adhesion to polymers, to high stresses in the deposited material; stress relaxation of the inorganic/polymer system then occurs. In our case, the wrinkling is visible in Figure 2b and leads to a considerable increase in the surface roughness, to 17.5 nm. The lateral dimensions (width) of the features of the gold layer deposited in these conditions are of the order 80–90 nm. A gold layer deposited in the same conditions directly onto a Si surface is smooth, with no visible structure, as noticed in SEM images.

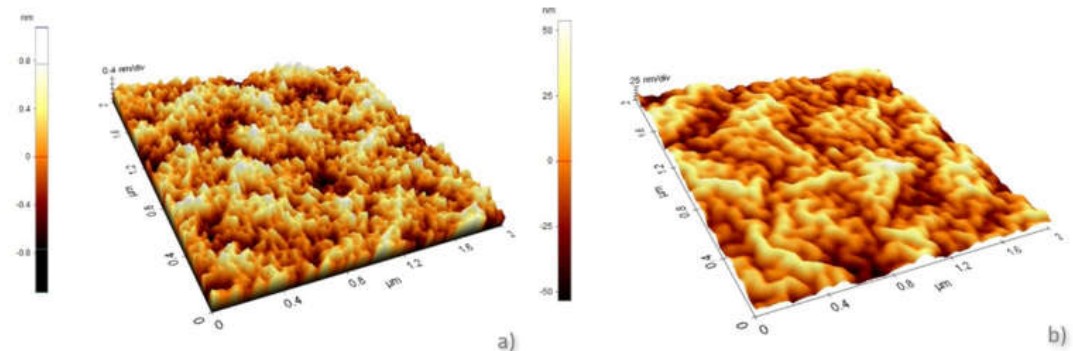

**Figure 2.** (**a**) AFM image of PMMA layer deposited on quartz; (**b**) AFM image of a gold layer deposited onto the PMMA/quartz surface in vacuum conditions (<$10^{-5}$ Torr). The gold layer was deposited using 2200 pulses.

Figure 3 presents the morphology of Au films deposited onto PMMA/quartz at different Ar pressures using different pulse numbers. At the lower pressure of 1 Torr and 10,000 ablation pulses, the film consists of nanoparticles with dimensions of the order of 20–30 nm, packed relatively densely, without any indication of coalescing into islands.

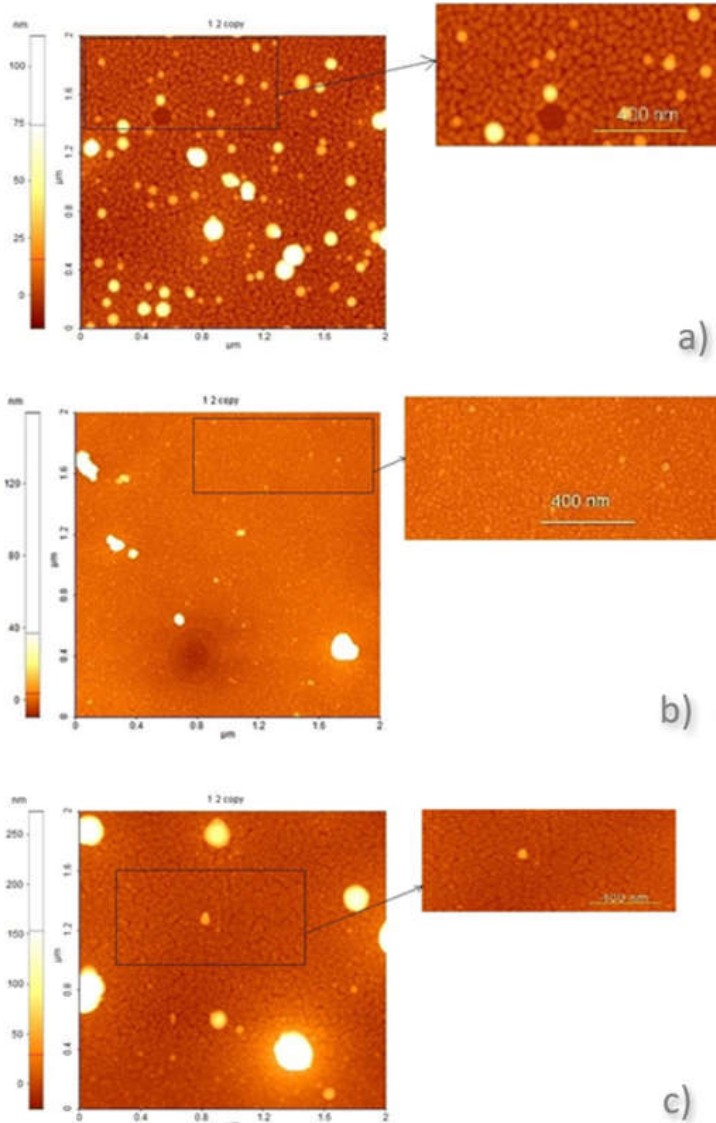

**Figure 3.** AFM images of Au layers deposited onto PMMA/quartz in (**a**) 1 Torr Ar, using 10,000 pulses (same as sensor S2); (**b**) 4 Torr, using 10,000 pulses (same as sensor S3); (**c**) 4 Torr Ar, using 30,000 pulses (same as sensor S4).

As the pressure increases to 4 Torr, also with 10,000 pulses, the dimensions of the nanoparticles are smaller—between 10 and 20 nm. The images are qualitatively similar for 1 Torr and for 4 Torr when 10,000 pulses are used, although the nanoparticle dimensions are different. However, when the number of ablation pulses is increased to 30,000 in 4 Torr Ar, the nanoparticles coalesce into islands of material separated by cracks having widths of about 13–20 nm, which make up about 59% of the surface of the gold film. The AFM image, in this case, is qualitatively different from those obtained for Au films deposited using 10,000 pulses, indicating a larger porosity of the film.

The histograms associated with the heights of the formations visible in AFM images can offer information on the differences between the morphologies of the gold films deposited in different conditions. If the large heights associated with the droplets visible in

images in Figure 3 are eliminated, the histograms of the images of films deposited with 10,000 pulses have widths of the same order of magnitude for 1 and 4 Torr Ar pressure, about 18 nm and 24 nm, respectively. In the case of the film deposited at 4 Torr using 30,000 pulses, however, the width of the histogram is much larger, namely, about 76 nm. This is another indication of the fact that films deposited at 4 Torr with 30,000 pulses are more porous than those deposited with 10,000 pulses; this will affect the properties of the sensors in which the films are used, as we will discuss below.

Figure 4 presents the morphology of Au films deposited in the same conditions onto PMMA/quartz and Si substrates in an Ar atmosphere. The difference between the Au films deposited in the same conditions onto the soft polymer surface and those deposited onto the hard Si surface is visible. At the relatively high gas pressures investigated, depositions onto the Si surface lead to islands of material separated by cracks, while depositions onto PMMA lead to less porous, more compact structures. At 1 Torr, 10,000 laser pulses lead to a gold film structure consisting of agglomerations of nanoparticles separated by cracks with an average width of 14 nm when a Si substrate is used (Figure 4a). The surface of the cracks makes up 39% of the total surface of the film, as estimated using the Image J software [23]. As described above, the same deposition conditions but onto a PMMA layer on top of a quartz substrate lead to more compact gold films consisting of nanoparticle agglomerations (Figure 4b). Gold films deposited onto Si substrates at a higher pressure of 4 Torr using 10,000 pulses (Figure 4c) have a morphology similar to the ones deposited at 1 Torr with 10,000 pulses. The surface of the cracks is about 42% of the total surface in this case, and their average width is about 10 nm. Again, deposition onto PMMA in the same conditions leads to relatively dense agglomerations of nanoparticles. In contrast, the films deposited onto Si at 4 Torr using 30,000 pulses are very porous, as can be seen in Figure 4e. In this case, the average width of the cracks, which make up 61% of the film, is 26 nm. Cross-sectional SEM images of the gold films deposited onto Si confirm its large porosity. These results are to be expected since it is well known that laser deposition at higher pressures leads to more porous films [16,24,25]. It is interesting to note that the morphology of the gold films deposited at 4 Torr using 30,000 pulses differs qualitatively from the ones deposited with 10,000 pulses (at 1 Torr and 4 Torr), both in the case of deposition onto the soft PMMA substrate and onto the hard Si one. The films deposited onto PMMA at 4 Torr with 30,000 pulses have cracks making up 59% of the film surface, similar to the case of films on silicon, but having smaller average width of about 16 nm. Thus, the gold surface consists, in this case, of a larger number of small cracks. Although deposition onto soft polymer surfaces leads to more compact gold films than those on hard silicon surfaces, the films deposited in 4 Torr Ar using 30,000 laser pulses are still the most porous ones, just as in the case of deposition onto silicon [16]. Therefore, qualitatively, films deposited onto soft polymer substrates have a morphology that is less favorable for sensors than those deposited onto hard substrates.

The AFM images of the surface after successive deposition of avidin and biotin onto the gold surface (Figure 5) are similar to those of the gold surface prior to deposition (Figure 3). The fact that the morphology of the surface does not change visibly after successive deposition of the proteins onto the gold indicates that the morphology of the biosensor surface is essentially given by the gold layer. The most porous surface (the one with the largest specific surface) is therefore still the one with the underlying Au surface deposited in 4 Torr Ar using 30,000 pulses. This proves the importance of the morphology of the gold layer in determining the characteristics of the sensor surface. The properties of the sensors are related to this morphology, as will be discussed below.

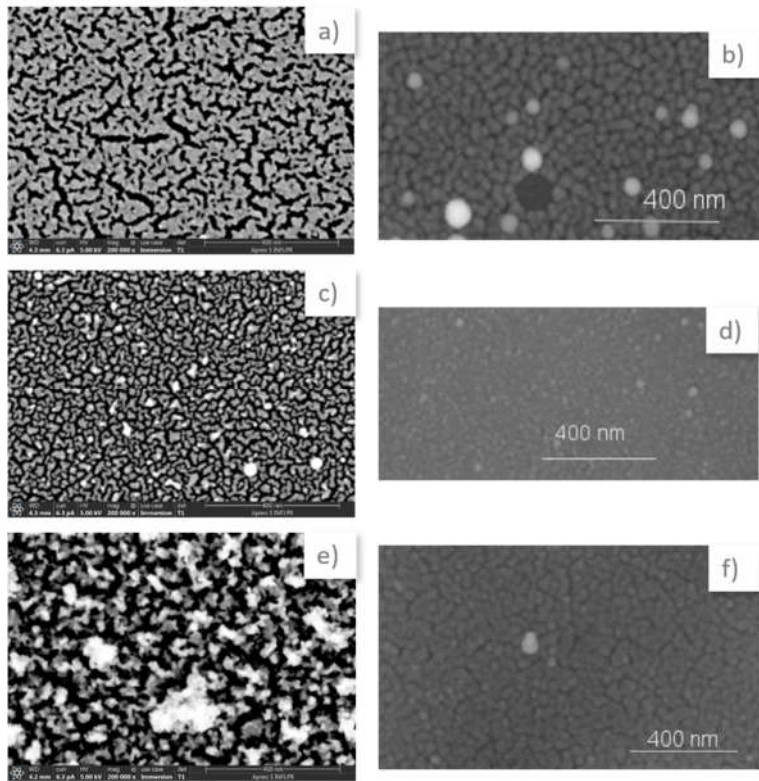

**Figure 4.** Difference in morphology of Au thin film deposited onto Si surface (left column) and PMMA surface (right column) in the same conditions: (**a**,**b**) 1 Torr; 10,000 p; (**c**,**d**) 4 Torr; 30,000 p; (**e**,**f**) 4 Torr; 30,000 p.

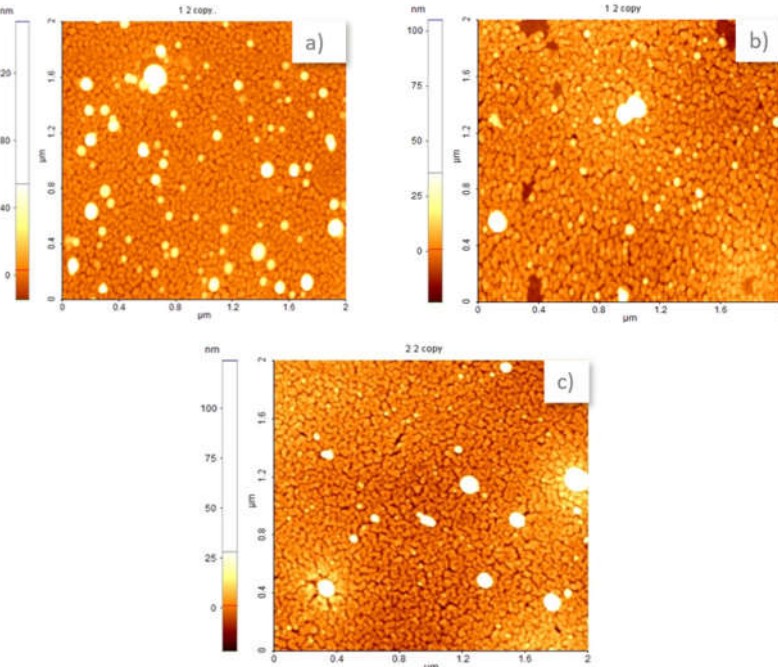

**Figure 5.** AFM images of the sensor surface after deposition of avidin or biotin/avidin on top of the Au layer: (**a**) avidin deposited on top of Au obtained in 1 Torr Ar using 10,000 p; (**b**) avidin deposited on top of Au obtained in 4 Torr Ar using 30,000 p; (**c**) biotin deposited on top of the avidin in (**c**).

### 3.2. Sensor Properties

Table 1 gives the properties of sensors S1 to S4, which have gold layers deposited in different conditions, as previously described. The oscillation frequencies after deposition of the successive layers were measured, and the frequency deviations caused by the depositions of the avidin on top of the gold and the biotin bound to avidin, respectively, were determined. The response of the sensors was measured by determining the initial oscillation frequency (before functionalization with avidin and biotin, respectively), and the frequency deviation from this after the 12 h incubation sequence, as described in the Materials and Methods Section. Since the surface of the avidin and biotin are washed to remove any unbound protein, the frequency changes of the sensors are an indication of the mass of protein that has been bound onto the underlying surface. The sensor with the dense gold layer deposited in vacuum (S1) clearly has the lowest sensitivity, having the smallest frequency changes. Sensors S2 and S3 have similar frequency changes, both after avidin deposition and after biotin deposition. This is probably related to the fact that their morphologies are qualitatively similar, as described above, based on the AFM images. Sensor S4 clearly has the best sensitivity, having the largest frequency changes. This is due to its larger porosity, which increases the effective surface of the sensitive layer. As discussed in relation to Figure 4, this is due to the morphology of the Au surface, which determines the avidin–biotin functionalized surface morphology.

**Table 1.** Oscillation frequencies of sensors S1–S4 after deposition of avidin and biotin layers on top of Au layers.

| Sensor | Ar Pressure. (Torr) | Number of Pulses | Oscillation Frequencies (MHz) | | Frequency Change (kHz) | Oscillation Frequencies (MHz) | Frequency Change (kHz) |
|---|---|---|---|---|---|---|---|
| | | | Au/PMMA | Avidin/ Au/PMMA | | Biotin/ Avidin/Au/PMMA | |
| S1 | $10^{-5}$ | 2200 | 68.468 | 68.463 | 5 | 68.459 | 4 |
| S2 | 1 | 10,000 | 68.776 | 68.761 | 15 | 68.749 | 12 |
| S3 | 4 | 10,000 | 68.256 | 68.239 | 17 | 68.227 | 12 |
| S4 | 4 | 30,000 | 68.884 | 68.848 | 36 | 68.829 | 19 |

We have observed that if we used a gold layer deposited with less than 2200 laser pulses, some areas of the Au layer were exfoliated during the immobilization process. On the other hand, using an Au layer deposited with more than 30,000 laser pulses, the attenuation of the surface wave is considered too high for the sensor to oscillate at the frequency for which it is designed (69 MHz) [20].

### 4. Discussion and Conclusions

Pulsed laser deposition at different pressures allows the morphology of the deposited gold film to be controlled. The presence of an ambient gas leads to a considerable decrease in the energy of the target species incident onto the substrate surface, due to hydrodynamic interactions with the ambient gas molecules [26]. The effect is more pronounced for a gas with a large mass of the species such as Ar and increases with increasing deposition pressure [18,26]. In particular, the study of the dynamics of laser-ablated Au species in an Ar atmosphere indicated that 4 cm away from the target, the energy of neutral Au species decreases by three orders of magnitude as the pressure increases from $10^{-5}$ Torr to 250 mTorr [18]. Such high laser deposition pressures also lead to films with a larger porosity [27]. Our previous research has shown that for deposition in 450 mTorr Ar, the morphology of the polymer surface is not affected by the impact of the gold species [16]. The morphology of the polymer surface is therefore not affected for deposition pressures larger than 450 mTorr, i.e., 1 Torr and 4 Torr, the conditions used for three of the SAW sensors we studied. We estimate that the energy of the gold species arriving onto the polymer surface is significantly below 1 eV, which is in the thermal domain [28]. However, although the morphology of the polymer surface is not modified by the incoming gold species in these relatively high-pressure conditions, the structure of the gold film deposited on top of

such soft surfaces differs considerably from that of gold films deposited onto hard surfaces such as Si or quartz [29]. This is due to the difference in mechanical properties between hard inorganic and soft polymer surfaces [29] and affects the sensing characteristics of the SAW biosensor.

In SERS-based sensors, wrinkling of the gold layer improves sensor properties [30]. In our case, the sensor based on a gold layer was deposited onto PMMA in a vacuum, which has a wrinkled morphology and poorer properties, compared to the sensors with nanoporous gold layers. In other cases studied (1 Torr and 4 Torr 10,000 pulses), Au layers on PMMA consist of a relatively dense agglomeration of NPs, which proves to be better than the wrinkled morphology obtained in a vacuum. The most porous layers, which are qualitatively different from the others we studied, are the ones deposited on the polymer using 30,000 pulses in 4 Torr Ar.

Physisorption of biotin-binding proteins was demonstrated to be a viable approach in biotin-binding functionality to gold surfaces [31]. Avidin is a tetrameric protein that can bind onto a wide range of surfaces, including gold layers. Moreover, previous works reported that proteins can accumulate on gold particles that are larger than 3 nm and form a stable adsorption layer [32]. Nevertheless, it is well known that Avidin is a tetrameric glycoprotein with four biotin-binding sites, with dimensions of $5.6 \times 5.0 \times 4.6$ nm. Whereas Avidin is a 66 kDa protein, Biotin is an organic heterobicyclic compound with a much smaller weight (0.244 kDaltons), which presents the major advantage for DNA binding and the high conjugating ability for many proteins and other molecules without significantly altering their biological activity [33]. The design of the PLD deposited porous gold interfaces for sensors aimed at an adequate size to allow proteins to attach to the surface and diffuse inside the material. Previous studies predicted that a pore dimension of 2 to 6 times the dimensions of a biomolecule (i.e., enzyme) is optimal for improving the biomolecule stability [34].

Given the established sizes of proteins, the designed width of the cracks on gold surfaces was in the range of 10–16 nm, therefore allowing optimal protein stability [34], and by varying the PLD conditions in increasing or decreasing the widths and the percentage of their presence on the surface, the sensors also could be adapted for larger proteins or biomolecules. There are clear changes in oscillation frequencies, evidencing the binding of avidin on surfaces results in cracks making up 59% of the film surface, with an average width of about 16 nm, which is almost 3 times bigger than protein size.

In our case, the AFM images show that both in the case of the surfaces functionalized with biotin and with biotin and avidin, the final surface morphologies closely follow that of gold, which proves the importance of controlling the gold morphology. High pressure and a large number of pulses lead to porous Au layers, which determine the morphology of the protein layers deposited on top of them. The layers deposited at 4 Torr using 30,000 pulses are therefore more favorable for the functionalization with biotin and avidin, as the sensor response indicates. Response of the sensor to avidin–biotin is a good indicator for the biosensor response to other organic materials since the biotin–avidin reaction is widely used in the detection of various biological materials [19]. The LW-SAW sensor with a porous Au layer is therefore promising for biosensing applications.

In conclusion, the properties of the LW-SAW biosensors are determined by the morphology of the laser-deposited Au layers that they contain, since the biotin and avidin layers subsequently deposited onto the gold surface reproduce its morphology. The properties of sensors with nanoporous gold layers deposited onto PMMA layers are better than the ones with dense layers. Therefore, the Au layers deposited at 1 and 4 Torr are more favorable for the functionalization with biotin and avidin because the final protein layers will closely follow its porous profile as well. Given the sizes of the proteins, the laser-deposited gold interfaces have optimal pore dimensions to ensure protein stability. The control of the morphology of the gold layer is thus important for the sensor properties and can be made by modifying the appropriate pulsed laser deposition conditions (high pressure, large number of pulses).

Our previous study [16] has demonstrated that LW-SAW sensors with porous gold layers can be successfully used for the detection of volatile organic compounds (VOCs). In the present case, the same LW-SAW design was used in a biosensor, the properties of which are characterized through the biotin–avidin reaction, a model reaction for biosensors. To the best of our knowledge, it is for the first time that the effect of Au layer nanoporosity on the sensing characteristics of LW-SAW biosensors to organic materials has been studied. In addition, control of pulsed laser deposition parameters through control of the dimensions of the cracks on the gold surfaces can also optimize the sensors for larger proteins or biomolecules.

**Author Contributions:** Conceptualization, C.V. and D.M.; formal analysis, C.V. and V.D.; investigation, C.V., D.M., V.D., and I.C.; writing—original draft preparation, D.M.; writing—review and editing, D.M. and C.V. All authors have read and agreed to the published version of the manuscript.

**Funding:** This work was supported by a grant from the Romanian Ministry for Research and Innovation, CCCDI-UEFISCDI, Project Nucleu 16N/08.02.2019.

**Institutional Review Board Statement:** Not applicable.

**Informed Consent Statement:** Not applicable.

**Data Availability Statement:** Not applicable.

**Acknowledgments:** The authors want to thank Antoniu Moldovan and Simona Brajnicov for AFM analysis.

**Conflicts of Interest:** The authors declare no conflict of interest.

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
