# Peer review of "Surface Acoustic Wave Biosensor with Laser-Deposited Gold Layer Having Controlled Porosity"

_chemosensors, doi:10.3390/chemosensors9070173_

Round 1

Reviewer 1 Report

Very clear and interesting paper. Only thing that I can recommend is to check for misprints, i.e. excessive hyphens (see lines 31, 32, 35, 38 etc.).

Author Response

Thank you for the review. We have re-checked the paper for misprints, and eliminated the excessive hyphens (which did not appear in the original paper submitted).

Reviewer 2 Report

A useful contribution to improving sensitivity of Love wave SAW biosensing. Some suggestions for the authors: 1. Provide how the SAWs were excited in the experimental methods section. 2. While results were obtained on PMMA guiding layer, and justified as a common material, it would have been interesting to include results on an oxide guiding layer, such as SiO2, considering that they AFM results showed significant improvements when PLD gold films were deposited on Si. 3. The effect of the porosity increase on insertion loss, if any, could be presented. 4. While there are very few readability issues with the English, many words were hyphenated, which should be corrected.

Author Response

Thank you for the review. Regarding the suggestions made, we have done the following:

  1. We have provided in the experimental methods section how the SAWs were excited:

“The input IDTs convert the electrical signal into a surface acoustic wave, and the output IDTs convert it back into an electrical signal. The changes in the electrical signal at detected at the output of the IDTs result from the interaction at the level of the surface of the sensors.”

“The frequency changes of the sensors were monitored using a frequency counter (Pendulum CNT-91, Spectracom Corp,Rochester, NY, USA) connected to a computer with Time View III software (Pendulum Instruments, Banino, Poland). A DHPVA-200 FEMTO amplifier (Messtechnik GmbH, Berlin, Germany) was used to compensate the signal loss from the oscillating circuit.”

  1. Thank you for the suggestion. We have already planned to undertake future research in which the sensors have oxide guiding layers instead of polymers, given that the SEM results for deposition onto Si indicated a higher porosity of the gold layers. We will present these results in a future paper.
  2. Higher porosity increases the attenuation of surface waves and the insertion loss, but in our case the thickness and porosity of the films used was chosen in such a way that, the insertion loss did not affect the oscillation of the sensors.

4. We have re-checked the paper for misprints, and eliminated the excessive hyphens (which did not appear in the original paper submitted).

Reviewer 3 Report

The point of this manuscript is that the nanoporous gold layer is helpful in increasing the sensitivity of the LW-SAW sensor. However, the efficacy of the nanoporous gold layer has been already verified in the Ref. 16. The difference of this manuscript from Ref. 16 is that the mass loading mechanism of the SAW sensor in this work is the binding between biotin and avidin molecules while that in Ref. 16 is simple physical adsorption of enzyme onto the nanoporous gold layer.

In that sense, in this work, much attention needs to be paid onto the change of the biotin-avidin coupling activity as a function of the nanoporous gold layer properties because the role of the nano-porous gold layer to provide a larger surface area for reaction has been proved already. However, this manuscript does not have any data on this issue and does not discuss this aspect at all. Without this information and discussion, this manuscript does not have clear novelty in comparison with Ref. 16.

The effectiveness of the nanoporous gold layer should be verified through well-defined and -controlled measurements with the SAW sensor. However, the manuscript does not show detailed experimental procedure, description of experimental environment, rigorous quantitative characterization of measurement variables, and efficacy of the present work. For example, the authors need to provide at least photographs of the LW-SAW sensor and experimental setup, detailed measurement conditions, and sample SAW sensor response data. They also need to show the reliability (repeatability) of the results in Table 1 and the noise level in the measure data. The structure and properties of the biotin and avidin molecules should be added, too, to better describe the mass loading effect. These are the minimum to support the credibility of the data in Table 1. The present manuscript describes only the change of morphology of the gold layer in terms of deposition conditions and that is it.

For the present, it is not easy to find the difference between this manuscript and Ref. 16, which impairs the novelty of this manuscript.

Author Response

Thank you for your comments and suggestions. We attached the answers.

Round 2

Reviewer 3 Report

The revised manuscript is better than the initial. However, it still needs more improvement.

The newly added paragraph at the bottom of P. 10 makes sense and helps improving the novelty of this manuscript. Abstract and the conclusion part at the bottom of p. 10 should be revised to incorporate this explanation as well.

In my previous review, I suggested “The effectiveness of the nanoporous gold layer should be verified through well-defined and -controlled measurements with the SAW sensor. However, the manuscript does not show any detailed experimental procedure, description of experimental environment, rigorous quantitative characterization of measurement variables, and efficacy of the present work. For example, the authors need to provide at least photographs of the LW-SAW sensor and experimental setup, detailed measurement conditions, and sample SAW sensor response data. They also need to show the reliability (repeatability) of the results in Table 1 and the noise level in the measure data. The structure and properties of the biotin and avidin molecules should be added, too, to better describe the mass loading effect. “.

However, in response to the comment, the authors have added only Fig. 1 with description of a very fundamental operation principle of a SAW sensor, which is not enough. Major revision is needed to address the above issue.

Author Response

In response to the comments in your first round of review, we attached a word document with a point by point response. It seems that you did not receive this word document and we apologize for this inconvenience. We are attaching the original word document with the point by point response, hoping that this time you will receive it. In order to be sure that you receive the information, we also give below the contents of this word document. Thank you for your patience.

Regarding your comment:

“The newly added paragraph at the bottom of P. 10 makes sense and helps improving the novelty of this manuscript. Abstract and the conclusion part at the bottom of p. 10 should be revised to incorporate this explanation as well.”

We have added revised the abstract, and the last two paragraphs of section 4 Results and Conclusions, as follows:

Abstract: : Laser-deposited gold immobilization layers having different porosities were incorporated into Love Wave Surface Acoustic Wave sensors (LW-SAWs). Variation of Pulsed Laser Deposition parameters allows good control of the gold film morphology. Biosensors with various gold film porosities were tested using the biotin-avidin reaction. Control of the Au layer morphology is important, since the biotin and avidin layer morphologies closely follow that of the gold. The response of the sensors to biotin/avidin, which is a good indicator of biosensor performance, is improved when the gold layer has increased porosity. Given the sizes of the proteins, the laser-deposited porous gold interfaces have optimal pore dimensions to ensure protein stability.

In conclusion, the properties of the LW-SAW biosensors are determined by the morphology of the laser-deposited Au layers that they contain, since the biotin and avidin layers subsequently deposited onto the gold surface reproduce its morphology. The properties of sensors with nanoporous gold layers deposited onto PMMA layers are better than the ones with dense layers. Therefore, the Au layers deposited at 1 and 4 Torr are more favorable for the functionalization with biotin and avidin because the final protein layers will closely follow its porous profile, as well. Given the sizes of the proteins, the laser-deposited gold interfaces have optimal pore dimensions to ensure protein stability. The control of the morphology of the gold layer is thus important for the sensor properties, and can be made by modifying the appropriate pulsed laser deposition conditions (high pressure, large number of pulses).

Our previous study [16] has demonstrated that LW-SAW sensors with porous gold layers can be successfully used for the detection of Volatile Organic Compounds (VOC). In the present case, the same LW-SAW design was used in a biosensor, the properties of which are characterized through the biotin-avidin reaction, a model reaction for biosensors. To the best of our knowledge, it is for the first time that the effect of Au layer nanoporosity on the sensing characteristics of LW-SAW biosensors to organic materials has been studied. In addition, control of pulsed laser deposition parameters, through control of the dimensions of the cracks on the gold surfaces, can also optimize the sensors for larger proteins or biomolecules.

In response to the comments in your first round of review, we attached a word document with a point by point response. It seems that you did not receive this word document and we apologize for this inconvenience. We are attaching the original word document with the point by point response, hoping that this time you will receive it. In order to be sure that you receive the information, we also give below the contents of this word document. Thank you for your patience.

Unfortunately, the system does not allow us to insert figures (the photograph of the sensor – figure 1, the photograph of the measurement system and the graph which illustrates the noise).

Unfortunately, the system does not allow us to insert figures (the photograph of the sensor – figure 1, the photograph of the measurement system and the graph which illustrates the noise).

Regarding requirement to include photographs of the LW-SAW sensor and experimental setup, a photograph of the LW-SAW sensor was included in the paper (figure 1):

“An image of the sensor is given in figure 1.”

Fig. 1 Photograph of the LW-SAW sensor mounted for measurement.

We provide here a photograph of the experimental setup, which is inside a Faraday cage in order to eliminate e.m. interferences.

In the picture, the frequency counter is on the right, the sensor chamber with 3 sensors is on the bottom left, the amplifier in the center left, and the computer for data acquisition on the top left.

We have added details on the experimental setup in the text:

“The frequency changes of the sensors were monitored using a frequency counter (Pendulum CNT-91, Spectracom Corp,Rochester, NY, USA) connected to a computer with Time View III software (Pendulum Instruments, Banino, Poland). A DHPVA-200 FEMTO amplifier (Messtechnik GmbH, Berlin, Germany) was used to compensate the signal loss from the oscillating circuit.”

We consider that, having added this text, the photograph of the experimental setup does not add significant information to this modified text, so that we have not added this photograph in the paper.

Regarding the requirement to provide detailed measurement conditions, this was added in the text:

“All measurements were made in a room where the temperature and humidity were kept constant at 29 °C and the humidity at 41%. The sensors were allowed to thermalize to room temperature, and then measurements were made.”

Regarding requirement to provide the sample SAW sensor response data, as we have added in the the first paragraph of results section 3.2. Sensor properties:

“The response of the sensors was measured by determining the initial oscillation frequency (before functionalization with avidin and biotin, respectively), and the frequency deviation from this after the 12 h incubation sequence, as described in the Materials and Methods section”

Since the measurement of the initial frequency and frequency after functionalization of the sensor surface were measured on consecutive days (after the 12 h incubation sequence), we cannot provide sample SAW sensor response data, as in the case of gas sensing, where the frequency shift can be visualized on the same readout.

Regarding the requirement to provide noise level, we have added the following information in the text:

“The noise level was measured in air (without analyte), after about 30 min operation of the sensor setup for thermalization. The noise level, estimated by measuring the frequency fluctuation over 10 min, represents the maximum frequency deviation from the trend line. Thus, a noise level between 120 Hz – 140 Hz was obtained for all the sensors. The noise level is much less than the frequency shift of the sensors (between 4-19 kHz.”

In the figure below is a sample SAW of noise level measured for the sensor S2. The figure below shows that the signal is stable after the first minute, and the noise level is less than 120 Hz.

Regarding the requirement to provide the structure and properties of the biotin and avidin molecules, the following information was provided in the text:

“Nevertheless, it is well known that Avidin is a tetrameric glycoprotein with four biotin-binding sites, with dimensions of 5.6 × 5.0 × 4.6 nm. Whereas Avidin is a 66 kDa protein, Biotin is an organic heterobicyclic compound with a much smaller weight (0. 244 kDaltons) which presents the major advantage for DNA binding and the high conjugating ability for many proteins and other molecules without significantly altering their biological activity [33].”

“The design of the PLD deposited porous gold interfaces for sensors aimed at an adequate size to allow proteins to attach not only to the surface, but to diffuse inside the material. Previous studies predicted that a pore dimension of 2 to 6 times the dimensions of a biomolecule (i.e. enzyme) is optimal for improving the biomolecule stability [34].

Given the established sizes of proteins, the designed width of the cracks on gold surfaces were in the range of 10-16 nm, therefore allowing an optimal protein stability [34], and by varying the PLD conditions in increasing or decreasing the widths and the percentage of their presence on the surface, the sensors also could be adapted for larger proteins or biomolecules. There are clear changes in oscillation frequencies, evidencing the binding of avidin on surfaces providing cracks making up 59% of the film surface, and an average width of about 16 nm, which is almost 3 times bigger than protein size”

Regarding the observation on the absence of novelty in the paper, in comparison to a previous one, the following elements of novelty are  present in the submitted paper:

  • In the previous work, which used a similar SAW sensor design, AChE was deposited on the nanoporous gold surface, and the sensor was used as a gas sensor and tested in Chloroform. In the present case the sensor was tested as a biosensor, using the avidin-biotin reaction, where the optimal conditions for biomolecule stability on nanoporous gold must be ensured:

“The design of the PLD deposited porous gold interfaces for sensors aimed an adequate size to allow proteins to attach not only to the surface, but to diffuse inside the material. Previous studies predicted that a pore dimension of 2 to 6 times the dimensions of a biomolecule (i.e. enzyme) is optimal for improving the biomolecule stability [33].”

  • The present paper presents additional results referring to the difference in morphology in the case of gold deposition onto soft substrates (PMMA) and onto hard ones (Si) which were not investigated before. In the previous paper, we characterized the gold films based on SEM images of gold films deposited onto Si; these films, however, have a different morphology from the gold films actually used in the sensors, which are deposited onto PMMA. In the present case, AFM microscopy was used to analyze the morphology of the films actually used in the sensors, including gold, avidin and biotin.
  • The ImageJ program was used to obtain additional quantitative information on the morphology of the surfaces, in order to better compare the films obtained in various conditions (not only based on qualitative comparison). Quantitative comparisons between the AFM images of the surface of the various layers making up the sensor were also obtained using the histograms associated with the heights of the formations visible in AFM images.

We mention that, due to an oversight for which we apologize, all comments on figure 2, including quantitative information on these images, was not included in the paper. We have added this text on page 4:

“Figure 3 presents the morphology of Au films deposited onto PMMA/quartz, at different Ar pressures, using different pulse numbers. At the lower pressure of 1 Torr and 10000 ablation pulses, the film consists in nanoparticles with dimensions of the order of 20 - 30 nm, packed relatively densely, without any indication of coalescing into islands. As the pressure increases to 4 Torr, also with 10000 pulses, the dimensions of the nanoparticles are smaller – between 10 and 20 nm.  The images are qualitatively similar for 1 Torr and for 4 Torr when 10000 pulses are used,  although the nanoparticle dimensions are different. However, when the number of ablation pulses is increased to 30000 in 4 Torr Ar, the nanoparticles coalesce into islands of material separated by cracks having widths of the about 13 – 20 nm, which make up about 59% of the surface of the gold film. The AFM image in this case is qualitatively different from those obtained for Au films deposited using 10000 pulses, indicating a larger porosity of the film.

The histograms associated with the heights of the formations visible in AFM images can offer information on the differences between the morphologies of the gold films deposited in different conditions. If the large heights associated with the droplets visible in images in figure 3 are eliminated, the histograms of the images of films deposited with 10000 pulses have widths of the same order of magnitude for 1 and 4 Torr Ar pressure, about 18 nm and 24 nm, respectively. In the case of the film deposited at 4 Torr using 30000 pulses, however, the width of the histogram is much larger, namely about 76 nm. This is another indication of the fact that films deposited at 4 Torr with 30000 pulses are more porous than those deposited with 10000 pulses; this will affect the properties of the sensors in which the films are used, as we will discuss below.” 

Round 3

Reviewer 3 Report

.